# Learning Active Learning from Data

**Ksenia Konyushkova**[*]
CVLab, EPFL
Lausanne, Switzerland
`ksenia.konyushkova@epfl.ch`

**Sznitman Raphael**
ARTORG Center, University of Bern
Bern, Switzerland
`raphael.sznitman@artorg.unibe.ch`

**Pascal Fua**
CVLab, EPFL
Lausanne, Switzerland
`pascal.fua@epfl.ch`

## Abstract

In this paper, we suggest a novel data-driven approach to active learning (AL). The key idea is to train a regressor that predicts the expected error reduction for a candidate sample in a particular learning state. By formulating the query selection procedure as a regression problem we are not restricted to working with existing AL heuristics; instead, we learn strategies based on experience from previous AL outcomes. We show that a strategy can be learnt either from simple synthetic 2D datasets or from a subset of domain-specific data. Our method yields strategies that work well on real data from a wide range of domains.

## 1 Introduction

Many modern machine learning techniques require large amounts of training data to reach their full potential. However, annotated data is hard and expensive to obtain, notably in specialized domains where only experts whose time is scarce and precious can provide reliable labels. Active learning (AL) aims to ease the data collection process by automatically deciding which instances an annotator should label to train an algorithm as quickly and effectively as possible.

Over the years many AL strategies have been developed for various classification tasks, without any one of them clearly outperforming others in all cases. Consequently, a number of meta-AL approaches have been proposed to automatically select the best strategy. Recent examples include bandit algorithms [2, 11, 3] and reinforcement learning approaches [5]. A common limitation of these methods is that they cannot go beyond combining pre-existing hand-designed heuristics. Besides, they require reliable assessment of the classification performance which is problematic because the annotated data is scarce. In this paper, we overcome these limitations thanks to two features of our approach. First, we look at a whole continuum of AL strategies instead of combinations of pre-specified heuristics. Second, we bypass the need to evaluate the classification quality from application-specific data because we rely on experience from previous tasks and can seamlessly transfer strategies to new domains.

More specifically, we formulate Learning Active Learning (LAL) as a regression problem. Given a trained classifier and its output for a specific sample without a label, we predict the reduction in generalization error that can be expected by adding the label to that datapoint. In practice, we show that we can train this regression function on synthetic data by using simple features, such as the variance of the classifier output or the predicted probability distribution over possible labels for a

---

[*]`http://ksenia.konyushkova.com`

specific datapoint. The features for the regression are not domain-specific and this enables to apply the regressor trained on synthetic data directly to other classification problems. Furthermore, if a sufficiently large annotated set can be provided initially, the regressor can be trained on it instead of on synthetic data. The resulting AL strategy is then tailored to the particular problem at hand. We show that LAL works well on real data from several different domains such as biomedical imaging, economics, molecular biology and high energy physics. This query selection strategy outperforms competing methods without requiring hand-crafted heuristics and at a comparatively low computational cost.

## 2 Related work

The extensive development of AL in the last decade has resulted in various strategies. They include uncertainty sampling [32, 15, 27, 34], query-by-committee [7, 13], expected model change [27, 30, 33], expected error or variance minimization [14, 9] and information gain [10]. Among these, uncertainty sampling is both simple and computationally efficient. This makes it one of the most popular strategies in real applications. In short, it suggests labeling samples that are the most uncertain, i.e., closest to the classifier's decision boundary. The above methods work very well in cases such as the ones depicted in the top row of Fig. 2, but often fail in the more difficult ones depicted in the bottom row [2].

Among AL methods, some cater to specific classifiers, such as those relying on Gaussian processes [16], or to specific applications, such as natural language processing [32, 25], sequence labeling tasks [28], visual recognition [21, 18], semantic segmentation [33], foreground-background segmentation [17], and preference learning [29, 22]. Moreover, various query strategies aim to maximize different performance metrics, as evidenced in the case of multi-class classification [27]. However, there is no one algorithm that consistently outperforms all others in all applications [28].

Meta-learning algorithms have been gaining in popularity in recent years [31, 26], but few of them tackle the problem of learning AL strategies. Baram et al. [2] combine several known heuristics with the help of a bandit algorithm. This is made possible by the maximum entropy criterion, which estimates the classification performance without labels. Hsu et al. [11] improve it by moving the focus from datasamples as arms to heuristics as arms in the bandit and use a new unbiased estimator of the test error. Chu and Lin [3] go further and transfer the bandit-learnt combination of AL heuristics between different tasks. Another approach is introduced by Ebert et al. [5]. It involves balancing exploration and exploitation in the choice of samples with a Markov decision process.

The two main limitations of these approaches are as follows. First, they are restricted to combining already existing techniques and second, their success depends on the ability to estimate the classification performance from scarce annotated data. The data-driven nature of LAL helps to overcome these limitations. Sec. 5 shows that it outperforms several baselines including those of Hsu et al. [11] and Kapoor et al. [16].

## 3 Towards data-driven active learning

In this section we briefly introduce the active leaning framework along with uncertainty sampling (US), the most frequently-used AL heuristic. Then, we motivate why a data-driven approach can improve AL strategies and how it can deal with the situations where US fails. We select US as a representative method because it is popular and widely applicable, however the behavior that we describe is typical for a wide range of AL strategies.

### 3.1 Active learning (AL)

Given a machine learning model and a pool of unlabeled data, the goal of AL is to select which data should be annotated in order to learn the model as quickly as possible. In practice, this means that instead of asking experts to annotate all the data, we select iteratively and adaptively which datapoints should be annotated next. In this paper we are interested in classifying datapoints from a target dataset $\mathcal{Z} = \{(x_1, y_1), \ldots, (x_N, y_N)\}$, where $x_i$ is a $D$-dimensional feature vector and $y_i \in \{0, 1\}$ is its binary label. We choose a probabilistic classifier $f$ that can be trained on some $\mathcal{L}_t \subset \mathcal{Z}$ to map

features to labels, $f_t(x_i) = \hat{y}_i$, through the predicted probability $p_t(y_i = y \mid x_i)$. The standard AL procedure unfolds as follows.

1. The algorithm starts with a small labeled training dataset $\mathcal{L}_t \subset \mathcal{Z}$ and large pool of unannotated data $\mathcal{U}_t = \mathcal{Z} \setminus \mathcal{L}_t$ with $t = 0$.

2. A classifier $f_t$ is trained using $\mathcal{L}_t$.

3. A query selection procedure picks an instance $x^* \in \mathcal{U}_t$ to be annotated at the next iteration.

4. $x^*$ is given a label $y^*$ by an oracle. The labeled and unlabeled sets are updated.

5. $t$ is incremented, and steps 2–5 iterate until the desired accuracy is achieved or the number of iterations has reached a predefined limit.

**Uncertainty sampling (US)**  US has been reported to be successful in numerous scenarios and settings and despite its simplicity, it often works remarkably well [32, 15, 27, 34, 17, 24]. It focuses its selection on samples which the current classifier is the least certain about. There are several definitions of maximum uncertainty but one of the most widely used ones is to select a sample $x^*$ that maximizes the entropy $\mathcal{H}$ over the probability of predicted classes:

$$x^* = \arg\max_{x_i \in \mathcal{U}_t} \mathcal{H}[p_t(y_i = y \mid x_i)] \,. \tag{1}$$

## 3.2 Success, failure, and motivation

We now motivate the need for LAL by presenting two toy examples. In the first one, US is empirically observed to be the best greedy approach, but in the second it makes suboptimal decisions. Let us consider simple two-dimensional datasets $\mathcal{Z}$ and $\mathcal{Z}'$ drawn from the same distribution with an equal number of points in each class (Fig. 1, left). The data in each class comes from a Gaussian distribution with a different mean and the same isotropic covariance. We can initialize the AL procedure of Sec. 3.1 with one sample from each class and its respective label: $\mathcal{L}_0 = \{(x_1, 0), (x_2, 1)\} \subset \mathcal{Z}$ and $\mathcal{U}_0 = \mathcal{Z} \setminus \mathcal{L}_0$. Here we train a simple logistic regression classifier $f$ on $\mathcal{L}_0$ and then test it on $\mathcal{Z}'$. If $|\mathcal{Z}'|$ is large, the test error can be considered as a good approximation of the generalization error: $\ell_0 = \sum_{(x', y') \in \mathcal{Z}'} \ell(\hat{y}, y')$, where $\hat{y} = f_0(x')$.

Let us try to label every point $x$ from $\mathcal{U}_0$ one by one, form a new labeled set $\mathcal{L}_x = \mathcal{L}_0 \cup (x, y)$ and check what error a new classifier $f_x$ yields on $\mathcal{Z}'$, that is, $\ell_x = \sum_{(x', y') \in \mathcal{Z}'} \ell(\hat{y}, y')$, where $\hat{y} = f_x(x')$. The difference between errors obtained with classifiers constructed on $\mathcal{L}_0$ and $\mathcal{L}_x$ indicates how much the addition of a new datapoint $x$ reduces the generalization error: $\delta_x = \ell_0 - \ell_x$. We plot $\delta_x$ for the 0/1 loss function, averaged over $10\,000$ experiments as a function of the predicted probability $p_0$ (Fig. 1, left). By design, US would select a datapoint with probability of class 0 close to 0.5. We observe that in this experiment, the datasample with $p_0$ closest to 0.5 is indeed the one that yields the greatest error reduction.

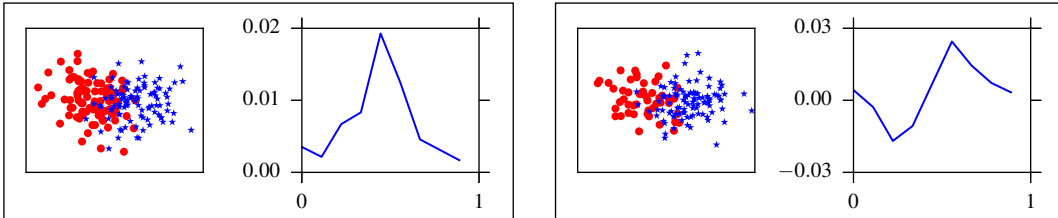

Figure 1: Balanced vs unbalanced. Left: two Gaussian clouds of the same size. Right: two Gaussian clouds with the class 0 twice bigger than class 1. The test error reduction as a function of predicted probability of class 0 in the respective datasets.

In the next experiment, the class 0 contains twice as many datapoints as the other class, see Fig. 1 (right). As before, we plot the average error reduction as a function of $p_0$. We observe this time that the value of $p_0$ that corresponds to the largest expected error reduction is different from 0.5 and thus the choice of US becomes suboptimal. Also, the reduction in error is no longer symmetric for the two classes. The more imbalanced the two classes are, the further from the optimum the choice made by

US is. In a complex realistic scenario, there are many other factors such as label noise, outliers and shape of distribution that further compound the problem.

Although query selection procedures can take into account statistical properties of the datasets and classifier, there is no simple way to foresee the influence of all possible factors. Thus, in this paper, we suggest Learning Active Learning (LAL). It uses properties of classifiers and data to predict the potential error reduction. We tackle the query selection problem by using a regression model; this perspective enables us to construct new AL strategies in a flexible way. For instance, in the example of Fig. 1 (right) we expect LAL to learn a model that automatically adapts its selection to the relative prevalence of the two classes without having to explicitly state such a rule. Moreover, having learnt the error reduction prediction function, we can seamlessly transfer LAL strategy to other domains with very little annotated data.

## 4   Monte-Carlo LAL

Our approach to AL is data-driven and can be formulated as a regression problem. Given a *representative* dataset with ground truth, we simulate an online learning procedure using a Monte-Carlo technique. We propose two versions of AL strategies that differ in the way how datasets for learning a regressor are constructed. When building the first one, LALINDEPENDENT, we incorporate unused labels individually and at random to retrain the classifier. Our goal is to correlate the change in test performance with the properties of the classifier and of newly added datapoint. To build the LALITERATIVE strategy, we further extend our method by a sequential procedure to account for selection bias caused by AL. We formalize our LAL procedures in the remainder of the section.

### 4.1   Independent LAL

Let the *representative* dataset[2] consist of a training set $\mathcal{D}$ and a testing set $\mathcal{D}'$. Let $f$ be a classifier with a given training procedure. We start collecting data for the regressor by splitting $\mathcal{D}$ into a labeled set $\mathcal{L}_\tau$ of size $\tau$ and an unlabeled set $\mathcal{U}_\tau$ containing the remaining points (Alg. 1 DATAMONTECARLO). We then train a classifier $f$ on $\mathcal{L}_\tau$, resulting in a function $f_\tau$ that we use to predict class labels for elements $x'$ from the test set $\mathcal{D}'$ and estimate the test classification loss $\ell_\tau$. We characterize the classifier state by $K$ parameters $\phi_\tau = \{\phi_\tau^1, \ldots, \phi_\tau^K\}$, which are specific to the particular classifier type and are sensitive to the change in the training set while being relatively invariant to the stochasticity of the optimization procedure. For example, they can be the parameters of the kernel function if $f$ is kernel-based, the average depths of the trees if $f$ is a tree-based method, or prediction variability if $f$ is an ensemble classifier. The above steps are summarized in lines 3–5 of Alg. 1.

---

**Algorithm 1** DATAMONTECARLO

1: **Input:** training set $\mathcal{D}$ and test set $\mathcal{D}'$, classification procedure $f$, partitioning function SPLIT, size $\tau$
2: **Initialize:** $\mathcal{L}_\tau, \mathcal{U}_\tau \leftarrow \text{SPLIT}(\mathcal{D}, \tau)$
3: train a classifier $f_\tau$
4: estimate the test set loss $\ell_\tau$
5: compute the classification state parameters $\phi \leftarrow \{\phi_\tau^1, \ldots, \phi_\tau^K\}$
6: **for** $m = 1$ **to** $M$ **do**
7:    select $x \in \mathcal{U}_\tau$ at random
8:    form a new labeled dataset $\mathcal{L}_x \leftarrow \mathcal{L}_\tau \cup \{x\}$
9:    compute the datapoint parameters $\psi \leftarrow \{\psi_x^1, \ldots, \psi_x^R\}$
10:    train a classifier $f_x$
11:    estimate the new test loss $\ell_x$
12:    compute the loss reduction $\delta_x \leftarrow \ell_\tau - \ell_x$
13:    $\xi_m \leftarrow \begin{bmatrix} \phi_\tau^1 & \cdots & \phi_\tau^K & \psi_x^1 & \cdots & \psi_x^R \end{bmatrix}, \delta_m \leftarrow \delta_x$
14: $\Xi \leftarrow \{\xi_m\}, \Delta \leftarrow \{\delta_m\} : 1 \leq m \leq M$
15: **Return:** matrix of learning states $\Xi \in \mathbb{R}^{M \times (K+R)}$, vector of reductions in error $\Delta \in \mathbb{R}^M$

| **Algorithm 2** BUILDLALINDEPENDENT | **Algorithm 3** BUILDLALITERATIVE |
|---|---|
| 1: **Input:** iteration range $\{\tau_{\min}, \ldots, \tau_{\max}\}$, classification procedure $f$ | 1: **Input:** iteration range $\{\tau_{\min}, \ldots, \tau_{\max}\}$, classification procedure $f$ |
| 2: SPLIT $\leftarrow$ random partitioning function | 2: SPLIT $\leftarrow$ random partitioning function |
| 3: **Initialize:** generate train set $\mathcal{D}$ and test dataset $\mathcal{D}'$ | 3: **Initialize:** generate train set $\mathcal{D}$ and test dataset $\mathcal{D}'$ |
| 4: **for** $\tau$ **in** $\{\tau_{\min}, \ldots, \tau_{\max}\}$ **do** | 4: **for** $\tau$ **in** $\{\tau_{\min}, \ldots, \tau_{\max}\}$ **do** |
| 5:    **for** $q = 1$ **to** $Q$ **do** | 5:    **for** $q = 1$ **to** $Q$ **do** |
| 6:      $\Xi_{\tau q}, \Delta_{\tau q} \leftarrow$ DATAMONTECARLO $(\mathcal{D}, \mathcal{D}', f, \text{SPLIT}, \tau)$ | 6:      $\Xi_{\tau q}, \Delta_{\tau q} \leftarrow$ DATAMONTECARLO $(\mathcal{D}, \mathcal{D}', f, \text{SPLIT}, \tau)$ |
| 7: $\Xi, \Delta \leftarrow \{\Xi_{\tau q}\}, \{\Delta_{\tau q}\}$ | 7:    $\Xi_\tau, \Delta_\tau \leftarrow \{\Xi_{\tau q}, \Delta_{\tau q}\}$ |
| 8: train a regressor $g : \xi \mapsto \delta$ on data $\Xi, \Delta$ | 8:    train regressor $g_\tau : \xi \mapsto \delta$ on $\Xi_\tau, \Delta_\tau$ |
| 9: construct LALINDEPENDENT $\mathcal{A}(g)$:     $x^* = \arg\max_{x \in \mathcal{U}_t} g[\xi_{t,x}]]$ | 9:    SPLIT $\leftarrow \mathcal{A}(g_\tau)$ |
| 10: **Return:** LALINDEPENDENT | 10: $\Xi, \Delta \leftarrow \{\Xi_\tau, \Delta_\tau\}$ |
|  | 11: train a regressor $g : \xi \mapsto \delta$ on $\Xi, \Delta$ |
|  | 12: construct LALITERATIVE $\mathcal{A}(g)$ |
|  | 13: **Return:** LALITERATIVE |

Next, we randomly select a new datapoint $x$ from $\mathcal{U}_\tau$ which is characterized by $R$ parameters $\psi_x = \{\psi_x^1, \ldots, \psi_x^R\}$. For example, they can include the predicted probability to belong to class $y$, the distance to the closest point in the dataset or the distance to the closest labeled point, but they do *not* include the features of $x$. We form a new labeled set $\mathcal{L}_x = \mathcal{L}_\tau \cup \{x\}$ and retrain $f$ (lines 7–13 of Alg. 1). The new classifier $f_x$ results in the test-set loss $\ell_x$. Finally, we record the difference between previous and new loss $\delta_x = \ell_\tau - \ell_x$ which is associated to the learning state in which it was received. The learning state is characterized by a vector $\xi_\tau^x = \begin{bmatrix} \phi_\tau^1 & \cdots & \phi_\tau^K & \psi_x^1 & \cdots & \psi_x^R \end{bmatrix} \in \mathbb{R}^{K+R}$, whose elements depend both on the state of the current classifier $f_\tau$ and on the datapoint $x$. To build an AL strategy LALINDEPENDENT we repeat the DATAMONTECARLO procedure for $Q$ different initializations $\mathcal{L}_\tau^1, \mathcal{L}_\tau^2, \ldots, \mathcal{L}_\tau^Q$ and $T$ various labeled subset sizes $\tau = 2, \ldots, T+1$ (Alg. 2 lines 4 and 5). For each initialization $q$ and iteration $\tau$, we sample $M$ different datapoints $x$ each of which yields classifier/datapoint state pairs with an associated reduction in error (Alg. 1, line 13). This results in a matrix $\Xi \in \mathbb{R}^{(QMT) \times (K+R)}$ of observations $\xi$ and a vector $\Delta \in \mathbb{R}^{QMT}$ of labels $\delta$ (Alg. 2, line 9).

Our insight is that observations $\xi$ should lie on a smooth manifold and that similar states of the classifier result in similar behaviors when annotating similar samples. From this, a regression function can predict the potential error reduction of annotating a specific sample in a given classifier state. Line 10 of the BUILDLALINDEPENDENT algorithm looks for a mapping $g : \xi \mapsto \delta$. This mapping is not specific to the dataset $\mathcal{D}$, and thus can be used to detect samples that promise the greatest increase in classifier performance in other target domains $\mathcal{Z}$. The resulting LALINDEPENDENT strategy greedily selects a datapoint with the highest potential error reduction at iteration $t$ by taking the maximum of the value predicted by the regressor $g$:

$$x^* = \arg\max_{x \in \mathcal{U}_t} g(\phi_t, \psi_x). \tag{2}$$

## 4.2 Iterative LAL

For any AL strategy at iteration $t > 0$, the labeled set $\mathcal{L}_t$ consists of samples selected at previous iterations, which is clearly *not* random. However, in Sec. 4.1 the dataset $\mathcal{D}$ is split into $\mathcal{L}_\tau$ and $\mathcal{U}_\tau$ randomly no matter how many labeled samples $\tau$ are available.

To account for this, we modify the approach of Section 4.1 in Alg. 3 BUILDLALITERATIVE. Instead of partitioning the dataset $\mathcal{D}$ into $\mathcal{L}_\tau$ and $\mathcal{U}_\tau$ randomly, we suggest simulating the AL procedure which selects datapoints according to the strategy learnt on the previously collected data (Alg. 3, line 10). It first learns a strategy $\mathcal{A}(g_2)$ based on a regression function $g_2$ which selects the most promising $3^{\text{rd}}$ datapoint when 2 random points are available. In the next iteration, it learns a strategy $\mathcal{A}(g_3)$ that selects $4^{\text{th}}$ datapoint given 2 random points and 1 selected by $\mathcal{A}(g_2)$ etc. In this way,

samples at each iteration depend on the samples at the previous iteration and the sampling bias of AL is represented in the data $\Xi, \Delta$ from which the final strategy LAL$_{\text{ITERATIVE}}$ is learnt.

The resulting strategies LAL$_{\text{INDEPENDENT}}$ and LAL$_{\text{ITERATIVE}}$ are both reasonably fast during the online steps of AL: they just require evaluating the RF regressor. The offline part, generating a datasets to learn a regression function, can induce a significant computational cost depending on the parameters of the algorithm. For this reason, LAL$_{\text{INDEPENDENT}}$ is preferred to LAL$_{\text{ITERATIVE}}$ when an application-specific strategy is needed.

## 5  Experiments

**Implementation details**   We test AL strategies in two possible settings: a) *cold start*, where we start with one sample from each of two classes and b) *warm start*, where a larger dataset of size $N_0 \ll N$ is available to train the initial classifier. In *cold start* we take the representative dataset to be a 2D synthetic dataset where class-conditional data distributions are Gaussian and we use the same LAL regressor in all 7 classification tasks. While we mostly concentrate on *cold start* scenario, we look at a few examples of *warm start* because we believe that it is largely overloooked in the litterature, but it has a significant practical interest. Learning a classifier for a real-life application with AL rarely starts from scratch, but a small initial annotated set is provided to understand if a learning-based approach is applicable at all. While a small set is good to provide an initial insight, a real working prototype still requires much more training data. In this situation, we can benefit from the available training data to learn a specialized AL strategy for an application.

In most of the experiments, we use Random Forest (RF) classifiers for $f$ and a RF regressor for $g$. The state of the learning process $\xi_t$ at time $t$ consists of the following features: a) predicted *probability* $p(y = 0 | \mathcal{L}_t, x)$; b) *proportion* of class 0 in $\mathcal{L}_t$; c) *out-of-bag* cross-validated accuracy of $f_t$; d) variance of *feature importances* of $f_t$; e) *forest variance* computed as variance of trees' predictions on $\mathcal{U}_t$; f) average *tree depth* of the forest; g) *size* of $\mathcal{L}_t$. For additional implementational details, including examples of the synthetic datasets, parameters of the data generation algorithm and features in the case of GP classification, we refer the reader to the supplementary material. The code is made available at `https://github.com/ksenia-konyushkova/LAL`.

**Baselines and protocol**   We consider the three versions of our approach: a) **LAL-independent-2D**, LAL$_{\text{INDEPENDENT}}$ strategy trained on a synthetic dataset of *cold start*; b) **LAL-iterative-2D**, LAL$_{\text{ITERATIVE}}$ strategy trained on a synthetic dataset of *cold start*; c) **LAL-independent-WS**, LAL$_{\text{INDEPENDENT}}$ strategy trained on *warm start* representative data. We compare them against the following 4 baselines: a) **Rs**, random sampling; b) **Us**, uncertainty sampling; c) **Kapoor** [16], an algorithm that balances exploration and exploitation by incorporating mean and variance estimation of the GP classifier; d) **ALBE** [11], a recent example of meta-AL that adaptively uses a combination of strategies, including **Us**, **Rs** and that of Huang et al. [12] (a strategy that uses the topology of the feature space in the query selection). The method of Hsu et al. [11] is chosen as a our main baseline because it is a recent example of meta AL and is known to outperform several benchmarks.

In all AL experiments we select samples from a training set and report the classification performance on an independent test set. We repeat each experiment 50–100 times with random permutations of training and testing splits and different initializations. Then we report the average test performance as a function of the number of labeled samples. The performance metrics are task-specific and include classification accuracy, IOU [6], dice score [8], AMS score [1], as well as area under the ROC curve (AUC).

### 5.1  Synthetic data

**Two-Gaussian-clouds experiments**   In this dataset we test our approach with two classifiers: RF and Gaussian Process classifier (GPC). Due to the the computational cost of GPC, it is only tested in this experiment. We generate 100 new unseen synthetic datasets of the form as shown in the top row of Fig. 2 and use them for testing AL strategies. In both cases the proposed LAL strategies select datapoints that help to construct better classifiers faster than **Rs**, **Us**, **Kapoor** and **ALBE**.

**XOR-like experiments**   XOR-like datasets are known to be challenging for many machine learning methods and AL is no exception. It was reported in Baram et al. [2] that various AL algorithms

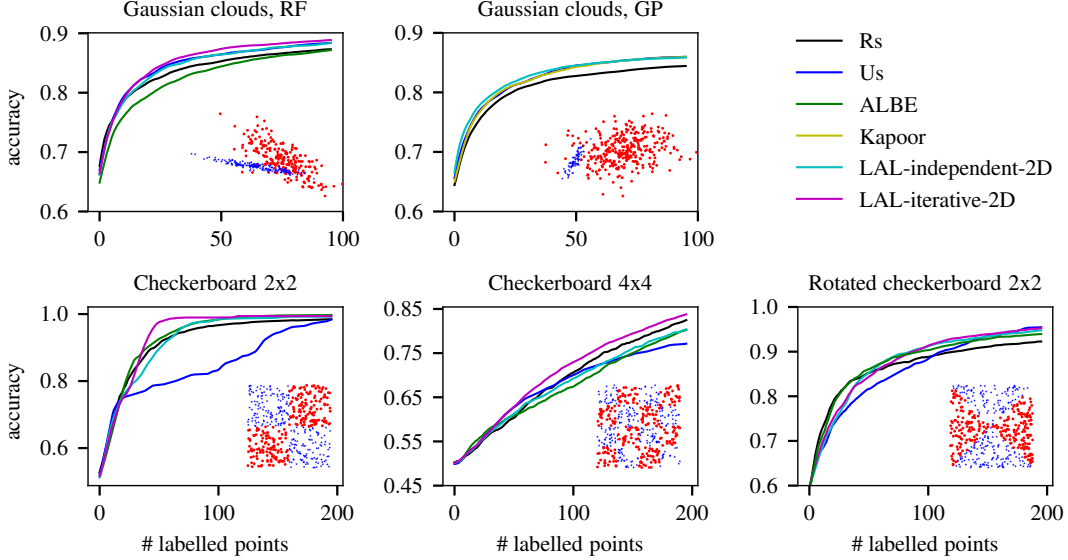

Figure 2: Experiments on the synthetic data. Top row: RF and GP on 2 Gaussian clouds. Bottom row from left to right: experiments on *Checkerboard* $2 \times 2$, *Checkerboard* $4 \times 4$, and *Rotated Checkerboard* $2 \times 2$ datasets.

struggle with tasks such as those depicted in the bottom row of Fig. 2, namely *Checkerboard* $2 \times 2$ and *Checkerboard* $4 \times 4$. Additionally, we consider *Rotated Checkerboard* $2 \times 2$ dataset (Fig. 2, bottom row, right). The task for RF becomes more difficult in this case because the discriminating features are no longer aligned to the axis. As previously observed [2], **Us** loses to **Rs** in these cases. **ALBE** does not suffer from such adversarial conditions as much as **Us**, but **LAL-iterative-2D** outperforms it on all XOR-like datasets.

## 5.2 Real data

We now turn to real data from domains where annotating is hard because it requires special training to do it correctly:

*Striatum*, 3D Electron Microscopy stack of rat neural tissue, the task is to detect and segment mitochondria [20, 17];

*MRI*, brain scans obtained from the BRATS competition [23], the task is to segment brain tumor in T1, T2, FLAIR, and post-Gadolinium T1 MR images;

*Credit card* [4], a dataset of credit card transactions made in 2013 by European cardholders, the task is to detect fraudulent transactions;

*Splice*, a molecular biology dataset with the task of detecting splice junctions in DNA sequences [19];

*Higgs*, a high energy physics dataset that contains measurements simulating the ATLAS experiment [1], the task is to detect the Higgs boson in the noise signal.

Additional details about the above datasets including sizes, dimensionalities and preprocessing techniques can be found in the supplementary materials.

**Cold Start AL**   Top row of Fig. 3 depicts the results of applying **Rs**, **Us**, **LAL-independent-2D**, and **LAL-iterative-2D** on the *Striatum, MRI*, and *Credit card* datasets. Both LAL strategies outperform **Us**, with **LAL-iterative-2D** being the best of the two. The best score of **Us** in these complex real-life tasks is reached 2.2–5 times faster by the **LAL-iterative-2D**. Considering that the LAL regressor was learned using a simple synthetic 2D dataset, it is remarkable that it works effectively on such complex and high-dimensional tasks. Due to the high computational cost of **ALBE**, we downsample *Striatum* and *MRI* datasets to 2000 datapoints (referred to as *Striatum mini* and *MRI mini*). Downsampling was not possible for the *Credit card* dataset due to the sparsity of positive labels (0.17%). We see in the bottom row of Fig. 3 that **ALBE** performs worse than

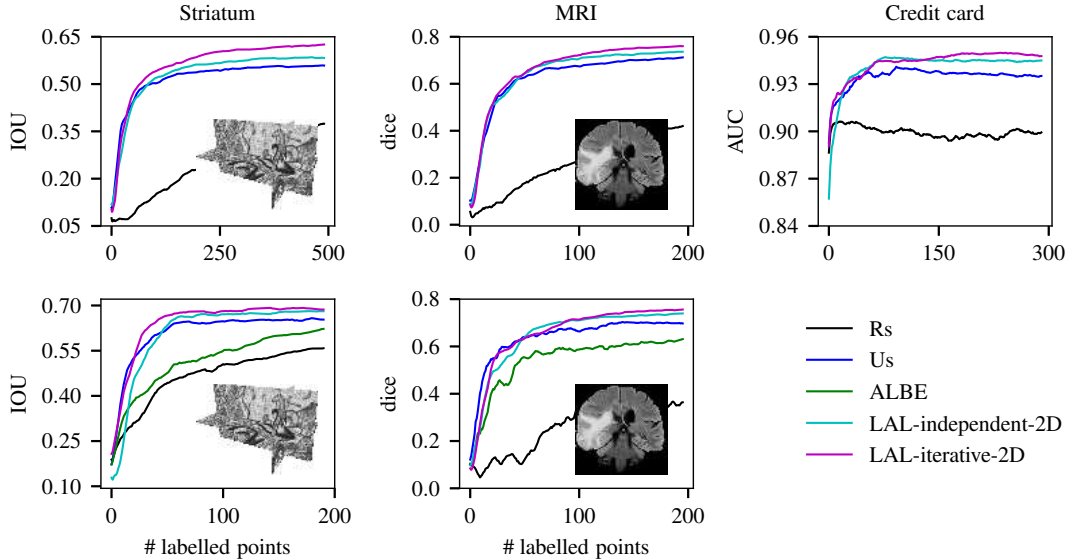

Figure 3: Experiments on real data. Top row: IOU for *Striatum*, dice score for *MRI* and AUC for *Credit card* as a function of a number of labeled points. Bottom row: Comparison with **ALBE** on the *Striatum mini* and *MRI mini* datasets.

**Us** but better than **Rs**. We ascribe this to the lack of labeled data, which **ALBE** needs to estimate classification accuracy (see Sec. 2).

**Warm Start AL**   In Fig. 4 we compare **LAL-independent-WS** on the *Splice* and *Higgs* datasets by initializing BUILDLALINDEPENDENT with 100 and 200 datapoints from the corresponding tasks. Notice that this is the only experiment where a significant amount of labelled data in the domain of interest is available prior to AL. We tested **ALBE** on the *Splice* dataset, however in the *Higgs* dataset the number of iterations in the experiment is too big. **LAL-independent-WS** outperforms other methods with **ALBE** delivering competitive performance—yet, at a high computational cost—only after many AL iterations.

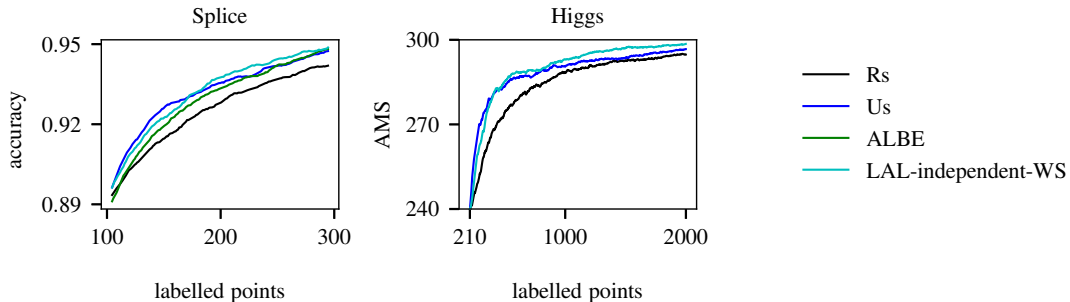

Figure 4: Experiments on the real datasets in warm start scenario. Accuracy for *Splice* is on the left, AMS score for *Higgs* is on the right.

## 5.3   Analysis of LAL strategies and time comparison

To better understand LAL strategies, we show in Fig. 5 (left) the relative importance of the features of the regressor $g$ for LALITERATIVE. We observe that both classifier state parameters and datapoint parameters influence the AL selection giving evidence that both of them are important for selecting a point to label. In order to understand what kind of selection LALINDEPENDENT and LALITERATIVE do, we record the predicted probability of the chosen datapoint $p(y^* = 0|\mathcal{D}_t, x^*)$ in 10 *cold start* experiments with the same initialization on the *MRI* dataset. Fig. 5 (right) shows the histograms of these probabilities for **Us**, **LAL-independent-2D** and **LAL-iterative-2D**. LAL strategies have

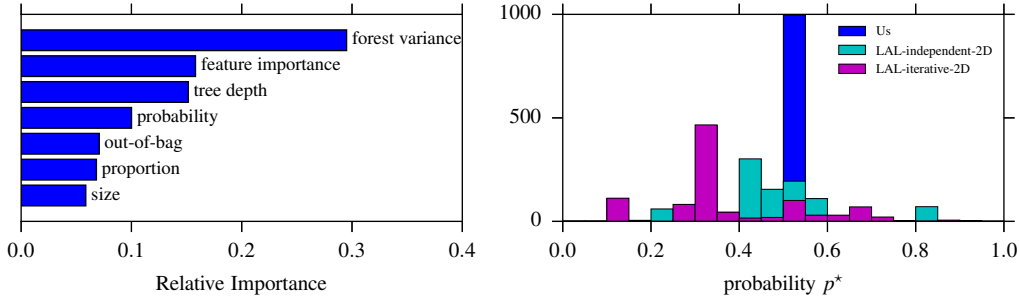

Figure 5: Left: feature importances of the RF regressor representing LALITERATIVE strategy. Right: histograms of the selected probability for different AL strategies in experiments with *MRI* dataset.

high variance and modes different from $0.5$. Not only does the selection by LAL strategies differ significantly from standard **Us**, but also the independent and iterative approaches differ from each other.

**Computational costs**   While collecting synthetic data can be slow, it must only be done *once, offline,* for all applications. Besides, Alg. 1, 2 and 3 can be trivially parallelised thanks to a number of independent loops. Collecting data offline for *warm start*, that is application specific, took us approximately $2.7$h and $1.9$h for *Higgs* and *Splice* datasets respectively. By contrast, the online user-interaction part is fast: it simply consists of learning $f_t$, extracting learning state parameters and evaluating the regressor $g$. The LAL run time depends on the parameters of the random forest regressor which are estimated via cross-validation (discussed in the supplementary materials). Run times of a Python-based implementation running on 1 core are given in Tab. 1 for a typical parameter set ($\pm$ 20% depending on exact parameter values). Real-time performance can be attained by parallelising and optimising the code, even in applications with large amounts of high-dimensional data.

Table 1: Time in seconds for one iteration of AL for various strategies and tasks.

| Dataset | Dimensions | # samples | Us | ALBE | LAL |
|---|---:|---:|---:|---:|---:|
| *Checkerboard* | 2 | 1000 | 0.11 | 13.12 | 0.54 |
| *MRI mini* | 188 | 2000 | 0.11 | 64.52 | 0.55 |
| *MRI* | 188 | 22 934 | 0.12 | — | 0.88 |
| *Striatum mini* | 272 | 2000 | 0.11 | 75.64 | 0.59 |
| *Striatum* | 272 | 276 130 | 2.05 | — | 19.50 |
| *Credit* | 30 | 142 404 | 0.43 | — | 4.73 |

## 6   Conclusion

In this paper we introduced a new approach to AL that is driven by data: Learning Active Learning. We found out that Learning Active Learning from simple 2D data generalizes remarkably well to challenging new domains. Learning from a subset of application-specific data further extends the applicability of our approach. Finally, LAL demonstrated robustness to the choice of type of classifier and features.

In future work we would like to address issues of multi-class classification and batch-mode AL. Also, we would like to experiment with training the LAL regressor to predict the change in various performance metrics and with different families of classifiers. Another interesting direction is to transfer a LAL strategy between different real datasets, for example, by training a regressor on multiple real datasets and evaluating its performance on unseen datasets. Finally, we would like to go beyond constructing greedy strategies by using reinforcement learning.

## Acknowledgements

This project has received funding from the European Union's Horizon 2020 Research and Innovation Programme under Grant Agreement No. 720270 (HBP SGA1). We would like to thank Carlos Becker and Helge Rhodin for their comments on the text, and Lucas Maystre for his discussions and attention to details.

## Footnotes

[2]The representative dataset is an annotated dataset that does not need to come from the domain of interest. In Sec. 5 we show that a simple synthetic dataset is sufficient for learning strategies that can be applied to various real tasks across various domains.

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
