[Supplementary Material · LALsup.pdf]

# Learning Active Learning from Data.
# Supplemental Materials

## A    Implementation details

**Representative dataset generation**    As a *representative* dataset in *cold start* experiments we use synthetic 2D datasets where each class comes from a Gaussian distribution with a randomly generated mean and variance. We set the size of training and test dataset to $400$ and $4000$ respectively and the proportion of class $0$ varies from $0.1$ to $0.9$. Each mean is drawn independently from a uniform distribution from $0$ to $1$ and the covariance is obtained by multiplying matrices whose entries are drawn uniformly between $-0.5$ and $0.5$ with their transposes. The LAL data generation parameters of Sec. 3 are set to the following values: $M = 100, T = 48, Q = 500$. For every new initialization we use a new representative dataset that insures that the learnt strategy can generalize to various problems.

In *warm start* experiments, we used 100 or 200 samples (in *Splice* and *Higgs* datasets correspondingly), out of which $40\%$ were used to estimate the test error and $60\%$ for collecting LAL data. Besides, we used multiple permutations of training and testing data to compensate for the limited amount of data (compared to the synthetic data). The LAL data generation parameters are the following. For *Splice* dataset, $Q = 100$ and $M = 10, \tau = 10, 14, \ldots, 48, T = 12$. For *Higgs* dataset, $Q = 100, M = 10$ and $\tau = 50, 55, \ldots, 110, T = 12$. The experiments show that is selected values are enough to interpolate between the learning states.

**Learning state parameters for GP**    When we use GP as a classifier, we operate on the following features: a) predicted probability $p(y = 0|\mathcal{D}_t, x)$ b) predicted variance by GP c) variance and d) lengthscale of RBF kernel e) kernel density estimation for $x$ with respect to labeled and f) unlabeles samples g) *size* of $L_t$.

**Cross-validation of LAL strategies**    The LAL regressor is represented by RF regressor that requires a set of meta-parameters. Their values were set with a cross validation of a regression problem with the regression performance is measured by R squared metrics. The cross-validated parameters for the LAL strategies can be found in a Tab 1

Table 1: Cross-validated parameters of LAL strategies

| Strategy | Dataset | # estimators | max depth of trees | max features per split |
|---|---|---|---|---|
| **LAL-independent-2D** | All | 2000 | 40 | 6 |
| **LAL-iterative-2D** | All | 1000 | 30 | 7 |
| **LAL-independent-WS** | *Splice* | 500 | 10 | 6 |
| **LAL-independent-WS** | *Higgs* | 1000 | 40 | 7 |

## B  Detailed descriptions of datasets

**2 Gaussian clouds**   When two Gaussian clouds datasets are used in AL experiments, they are generated with the same procedure as for the representative dataset in *cold start* (see Sec. A). Parameters of the data generation process are set at random every time, thus these datasets are not seen by LAL. A few examples of these daatsets are depicted in Fig. 1

Figure 1: 4 examples of the synthetic datasets with as a representative dataset and in the experiments with 2 Gaussian clouds.

**Striatum**   This dataset consists of 3D Electron Microscopy stack of rat neural tissue from striatum [5, 3] (Fig. 2). The train stack is of size $318 \times 711 \times 422$ pixels and the test stack is of size $318 \times 711 \times 450$ with the resolution of 5nm in all three spatial orientations. The task is to detect and segment mitochondria – intracellular structures that supply the cell with its energy. It is a laborious task for neuroscientists to annotate sufficient amounts of data to learn a classifier. Furthermore, the visual appearance varies significantly for different areas in the brain, for different animal species and for different settings of the equipment. The images are oversegmented with [1] and features are extracted according to Lucchi et al. [5]. The properties of the resulting dataset are summarized in Tab. 2

(yz)  (xy)

volume cut  (xz)

Figure 2: Interface of the FIJI Visualization API, which is extensively used to interact with 3D image stacks. The user is presented with three orthogonal planar slices of the stack. While effective when working slice by slice, this is extremely cumbersome for random access to voxels anywhere in the 3D stack, which is what a naive AL implementation would require.

Table 2: Parameters of the datasets.

| Dataset | Dimensions | # training samples | # test samples | positive class % |
|---------|-----------:|-------------------:|---------------:|-----------------:|
| 2 Gauss clouds | 2 | 400 | 4000 | 50 |
| Checkerboard | 2 | 1000 | 1000 | 50 |
| Striatum | 272 | 276 130 | 294 496 | 11.59 |
| Striatum mini | 272 | 2000 | 2000 | 11.59 |
| MRI | 188 | 22 934 | 22 562 | 5.99 |
| MRI mini | 188 | 2000 | 2000 | 5.99 |
| Credit | 30 | 142 403 | 142 404 | 0.17 |
| Splice | 60 | 1000 | 2175 | 48.09 |
| Higgs | 30 | 125 000 | 125 000 | 34.26 |

Figure 3: bla bla

**MRI**    20 MRI brain scans of Fig. 3 are obtained from BRATS competition [6]. The task is to segment brain tumor in T1, T2, FLAIR, and post-Gadolinium T1 MR images. We follow the protocol similar to the described in *Striatum* and oversegment stacks first and then extract feature with the convolutions of images with standard filters such as Gaussian, gradient filter, tensor, Laplacian of Gaussian and Hessian with different parameters. Remember that different permutations of training and testing data are used in AL experiments in order to better assess the classification quality. However, in imaging domain the samples (pixels) are not independent. This in *MRI* we permute the whole scans of different patients and in *Striatum* the size of the test stack is big enough ( 300 000 samples) to evaluate prediction quality accurately.

**Credit card**    The task is to detect credit card fraud transactions in transaction made by European cardholders in September 2013 [2]. The obtained 30 features are the result of PCA on the real features that are not provided due to the confidentiality issues. This is highly imbalanced dataset with only 0.17% of fraud transactions among normal transactions (see Tab. 2).

**Splice**    In this dataset from the domain of molecular biology, our task is to detect splice junctions between exons and introns in DNA sequences [4]. The sequences attributes are encoded numerically and a problem is formulated as a binary classification task.

**Higgs**    This dataset from the domain of high energy physics contains the data that simulates the ATLAS experiment [**?** ]. Higgs Boson detection challenge has its task to classify events into classes of tau tau decay of a Higgs boson and background noise. We preprocess the data by replacing missing feature values with the median of the corresponding feature.

# C  Additional experimental results

 Due to the space constraints, figures in the main manuscript are small and hard to see. Thus, we show
 them again here with a higher resolution. Moreover, we present experiments with additional quality
 measures that couldn't fit in the main paper.

Figure 4: Experiments on synthetic data. 2 Gaussian clouds, RF classifier.

Figure 5: Experiments on synthetic data. 2 Gaussian clouds, GP classifier.

Figure 6: Experiments on synthetic data. XOR-like dataset, *Checkerboard* $4 \times 4$.

Figure 7: Experiments on synthetic data. XOR-like dataset, *Checkerboard* $2 \times 2$.

Figure 8: Experiments on synthetic data. XOR-like dataset, *Banana*.

Figure 9: Experiments on real data with cold start, *Striatum*.

Figure 10: Experiments on real data with cold start, *Striatum mini*.

Figure 11: Experiments on real data with cold start, *MRI*.

Figure 12: Experiments on real data with cold start, *MRI mini*.

Figure 13: Experiments on real data with cold start, *Credit card*.

Figure 14: Experiments on real data with warm start, accuracy measure on *Splice*.

Figure 15: Experiments on real data with warm start, AUC measure on *Splice*.

Figure 16: Experiments on real data with warm start, accuracy measure on *Higgs*.

Figure 17: Experiments on real data with warm start, AUC measure on *Higgs*.