[Reviews · NeurIPS 2017]

Reviewer 1



Key insights/novelty: * Trains a regression model that predicts improvement in generalisation error * To the best of my knowledge this is an original approach to this problem (although similar, but sufficiently different, methods do exist) Questions: * In the experimental section, I recommend that the authors also include experiments on rotated checkerboards - the discriminating features of checkerboards are aligned to the axes, and using random forest classifiers may yield an overly optimistic performance curve. Rotating at 45 degrees may yield overly pessimistic curve, but it will explain the utility of the proposed methods better. * Baseline methods are not satisfactory since none of the proposed baselines can learn as much as the topology of the feature space as the proposed method (eg Zhu, Ghahramani, Lafferty ICML2003). At the very least this method should be compared in the discussions. * I'm not convinced by the results since they don't seem to be significantly better than baselines. Particularly with uncertainty sampling on the real data. This point relates to my suggestion of rotating the checkerboard so that the discriminating features will not be axis-aligned. * This method will require a relatively substantial proportion of data to be labelled to begin with. While you say that only a couple of hours were spent in labeling for the offline experiments, this time is domain-specific. Are there any techniques that can be used to improve this? * The conclusions of this paper were very short and underwhelming. General Statements: * Very well written, clear motivation. Easy to follow story. Small typos/recommended corrections: * L79. Unannotated data? * Showing the relative accuracy only shows one side of the picture. It would be insightful to show the accuracies of both too. Practitioners will be interested in knowing whether it is normally with learning strategies that have a good initial generalisation that disimprove moving forward. * Algorithm 1: unclear which datasets the classifiers are trained on. Recommend clarifying. And in line 14, please clarify. Do you mean $\{\delta_m: 1 \leq m \leq M\}$? * Should a selection function be passed into algorithm 1 to clarify line 7?

Reviewer 2



This paper proposes a novel active learning method, which trains a regression model to predict the expected error reduction for a candidate example in a particular learning state. The studied problem is very interesting and novel. Compared to existing active learning approaches with hand-crafted heuristics, the proposed approach can learn the selection strategies based on the information of the data example and model status. The experiments show that this method works well both in synthetic data and real data. At the early state of active learning, there is very few labeled data. That implies there is very few training data for the regressor, and the training data itself is not reliable. So the prediction of the regressor could be inaccurate, i.e., the predicted expected error reduction may be inaccurate, resulting less effective sample selection. The learning process has Q different initializations, T various labeled subset sizes and M data points in each iteration, so the matrix of observations is QMT*(K+R). If these three numbers are large, the learning process could be time consuming. There are many state-of-the-art active learning methods. It would be more convincing if more methods compared in the experiments.

Reviewer 3



This paper proposes a data-driven method for active learning, using a regressor to select the best strategy. The paper is well-organized, however, there are several concerns about the paper: 1. The contribution of this paper is not significant. The proposed method has limitation, it requires previous datasets and there is no guarantee about the performance of the warm start. Without filling the gap between the optimal regressor and the one created by Monte-Carlo approach, the experiment results are not convincing. 2. The proposed method is too heuristic, there is no theoretical foundation. 3. What is the representative dataset? It needs to be explained clearly. 4. Why did the authors not use the UCI datasets in [11]? The authors should provide some explanations about the choice of the datasets.